# Player Chronotype Does Not Affect In-Game Performance during the Evening (>18:00 h) in Professional Male Basketball Players

Michael Pengelly [1],* , Nathan Elsworthy [1] , Joshua Guy [1], Aaron Scanlan [2] and Michele Lastella [3]

1 School of Health, Medical and Applied Science, Central Queensland University, Cairns, QLD 4870, Australia; n.elsworthy@cqu.edu.au (N.E.); j.guy@cqu.edu.au (J.G.)
2 Human Exercise and Training Laboratory, Central Queensland University, Rockhampton, QLD 4701, Australia; a.scanlan@cqu.edu.au
3 Appleton Institute for Behavioural Science, Central Queensland University, Adelaide, SA 5034, Australia; m.lastella@cqu.edu.au
* Correspondence: michael.pengelly@cqumail.com; Tel.: +61-490871709

**Abstract:** Sport-specific skills display diurnal variation across various team sports such as badminton and tennis serving accuracy and soccer dribbling, volleying, and chipping execution. However, the effects of athlete chronotype on in-game sport-specific skill performance according to time of day across team sports is not well understood. Therefore, the aim of this study was to identify the effect of player chronotype on in-game basketball performance during evening games. Professional male basketball players ($n = 11$) completed a morningness–eveningness questionnaire and were categorized according to chronotype (morning-type: $n = 4$; neither-type: $n = 6$; evening-type: $n = 1$). Box score data from the 2019/20 season were utilized to determine individual in-game performance during evening games played after 18:00 h. Composite metrics (i.e., effective field goal percentage, offensive rating, defensive rating, and player efficiency) were used as indicators of player performance. Non-significant ($p \geq 0.21$) differences were evident between M-types and N-types for most performance measures. Small to very large effects were observed in the number of rebounds favoring M-types, and three-point shots attempted and made, assists, and steals favored N-types. In-game performance appeared to not be affected by chronotype (i.e., M-type vs. N-type) in evening games among professional male basketball players. The lack of observed effect between chronotype and in-game performance suggest coaching staff may not need to consider player chronotype when developing a match strategy or assigning player roles if largely dealing with M-types and N-types. However, to ensure the greatest specificity, coaching staff may endeavor to schedule habitual training times in line with that of competition in an effort to align player circadian rhythms to games.

**Keywords:** morningness; eveningness; diurnal variation; team sport; game performance; circadian rhythm

## 1. Introduction

Daily circadian rhythmical oscillations occur in several physiological and behavioral functions that contribute to athletic performance (e.g., body temperature and cortisol concentration) [1–3]. Previous studies report a clear circadian rhythm in sports performance with oxygen uptake, peak power, and mean power during a 30 s Wingate test [4], isometric knee extensor strength [5], and aerobic endurance during a 20 m multistage shuttle run test [6] all being affected by the time of day. In this sense, the acrophase of body temperature has been identified to be concurrent with peaks in physical capacities such as muscular strength [5] and anaerobic power [4,7], which peak in the evening. The later acrophase of physical capacities in the evening is therefore considered the primary factor underpinning the consistent finding that sports performance typically peaks in the evening [1,2,8–11].

This consensus extends to sport-specific skills such as soccer volleying, chipping, and dribbling execution [1,12]; badminton serve accuracy [11]; and tennis serve velocity [9,10], demonstrating diurnal variations favoring afternoon (14:00–16:00 h) and evening (19:00–21:00 h) performance compared to morning (07:00–09:00 h).

While diurnal variations may exist among sport-specific skills, previous studies have failed to report differences across the day, specifically according to player chronotypes [1,9–11]. Further, there remains limited data examining the effect of chronotype on in-game sport-specific skill performance according to the time of day. The potential effect of chronotype according to time of day is of particular importance, as athletes tend to choose, pursue, and excel in sports aligning to their chronotype with a predominance of neither-types (N-types) exhibited among team sport athletes who are often subjected to games scheduled in the evening for their respective sport [13,14], that is, athletes who display neither a preference for morningness nor eveningness [15,16]. Winter et al. [17] offer preliminary findings examining the effect of chronotype on batting performance among professional baseball players indicating that morning-types (M-types) had a higher batting average in early afternoon games (<14:00 h), while evening-types (E-types) had a higher batting average during evening games (>20:00 h). These data emphasize the need for greater research pertaining to the effects of chronotype on sport-specific performance during games.

Basketball is one sport where diurnal variation in sport-specific skills may be expected with key physical movements performed during games in multi-directional paths at varying velocities [18]. For instance, basketball players undergo repeated bouts of walking, running, shuffling, jumping, and actions requiring the arms to be extended overhead such as for shooting and rebounding throughout games [18]. In this sense, the physical movements performed during basketball games are dependent on players possessing adequate strength, power, anaerobic capacity, and aerobic capacity [18]. In turn, strength [5], power [1], anaerobic capacity [19], and aerobic capacity [6] have been shown to vary across the day which, in turn, may affect individual performance in basketball. However, these physical attributes are not indicative of game outcome or in-game performance in basketball and, therefore, other measures require consideration in chronotype studies.

Shooting performance has previously been demonstrated to be positively affected by sleep, with Mah et al. [20] reporting free-throw shooting percentage and three-point field goal percentages to each increase by ~9% following encouragement to obtain as much nocturnal sleep as possible (with a minimum of 10 h in bed each night) among male collegiate division one basketball players. In addition, the association between sleep measures and in-game performance measures has been established in basketball by Fox et al. [21], who identified positive associations between subjective sleep quality and offensive rating ($\beta = 8.59$, $p = 0.02$) as well as player efficiency ($\beta = 5.49$, $p = 0.01$). Sport-specific skills, such as shooting performance, however, are yet to be examined in relation to chronotype. It is plausible that if shooting performance improves with prolonged sleep (~10 h), in-game execution of sport-specific skills may also be affected with the scheduling of games. In this way, later start times would favor E-types as peak body temperature (~19:00 h) [1,22], onset of melatonin (24:36 h), and daytime sleepiness (5–9 h) occur later in the day compared to M-types [23]. Equally, evening games (18:00–20:30 h) for M-types may contribute to a lower shooting percentages if players are not aroused due to the earlier onset of melatonin in daytime sleepiness compared to E-types [23]. With performance measures, such as shooting percentage, indicative of match outcomes [24] and in-game performances in basketball [25], it is pertinent to establish whether this type of performance is affected by chronotype as it is by sleep [20]. Therefore, the aim of this study was to identify the effect of evening games (>18:00 h) on basketball performance between M-types, N-types, and E-types.

## 2. Results

The median and IQR for all in-game performance measures in each chronotype group are shown in Table 1. A small, significant effect of chronotype was observed for blocked shots with M-types registering more blocks per game than N-types ($p < 0.05$). No significant ($p > 0.05$) differences among the remaining in-game performance measures were apparent between M-types and N-types. However, effect size analyses showed N-types registered more made three-point shots ($-1.98$; large effect), attempted three-point shots ($-1.86$, large effect), assists ($-1.42$; large effect), and steals ($-3.03$; very large effect) during evening games compared to M-types (Table 1). Low statistical power (<0.80) was associated with most in-game performance measures except steals (0.98; Table A1).

**Table 1.** Median (inter-quartile range) values mean (standard deviation) and comparison statistics between morning-types (M-types) and neither-types (N-types) for in-game performance measures during evening games in professional male basketball players.

| Measure | M-Types (*n* = 4) | | N-Types (*n* = 6) | | | |
|---|---|---|---|---|---|---|
| | Median (IQR) | Mean (SD) | Median (IQR) | Mean (SD) | Cohen's d (95% CIs) | *p*-Value |
| Minutes | 17.8 (14.1–27.4) | 20.7 (8.0) | 26.5 (18.9–32.1) | 25.4 (7.9) | −0.95 (−2.18 to 0.46) | 0.52 |
| 2-Points made | 3.0 (1.0–5.0) | 3.1 (2.5) | 2.0 (1.0–4.0) | 2.3 (2.0) | 0.28 (−1.01 to 1.53) | 0.85 |
| 2-Points attempted | 5.0 (2.0–7.0) | 5.1 (3.6) | 4.0 (2.0–6.0) | 4.2 (3.1) | 0.36 (−0.95 to 1.60) | 0.81 |
| 3-Points made | 0.0 (0.0–1.0) | 0.8 (1.1) | 1.0 (1.0–2.0) | 1.5 (1.3) | −1.98 (−3.28 to −0.30) | 0.21 |
| 3-Points attempted | 2.0 (0.0–4.0) | 2.3 (2.2) | 4.0 (2.0–6.0) | 4.0 (2.3) | −1.86 (−3.14 to −0.21) | 0.24 |
| Points | 8.0 (4.3–14.8) | 10.0 (7.4) | 10.5 (5.0–16.0) | 11.2 (7.3) | −0.77 (−2.00 to 0.60) | 0.61 |
| Effective field goal percentage | 56.4 (37.5–71.4) | 54.1 (29.4) | 54.6 (41.1–69.2) | 54.8 (26.6) | −0.38 (−1.62 to 0.93) | 0.80 |
| Assists | 1.0 (0.0–2.0) | 1.1 (1.2) | 1.0 (0.0–3.0) | 1.7 (1.6) | −1.42 (−2.67 to 0.10) | 0.36 |
| Rebounds | 4.0 (2.0–7.0) | 5.2 (4.5) | 3.0 (2.0–5.0) | 3.8 (2.4) | 0.74 (−0.63 to 1.97) | 0.62 |
| Steals | 0.0 (0.0–1.0) | 0.4 (0.6) | 1.0 (0.0–1.0) | 0.8 (1.0) | −3.03 (−4.48 to −0.99) | 0.70 |
| Blocks | 0.0 (0.0–1.0) | 0.7 (1.1) | 0.0 (0.0–0.0) | 0.3 (0.7) | 0.4 (−0.91 to 1.64) | 0.04 * |
| Offensive rating per minute | 0.8 (0.6–1.0) | 0.8 (0.3) | 0.7 (0.5–0.9) | 0.7 (0.3) | −0.29 (−1.54 to 1.01) | 0.85 |
| Defensive rating per minute | 0.3 (0.2–0.3) | 0.3 (0.1) | 0.2 (0.2–0.3) | 0.2 (0.1) | 0.67 (−0.69 to 1.89) | 0.60 |
| Player efficiency per minute | 0.5 (0.3–0.7) | 0.5 (0.3) | 0.4 (0.3–0.6) | 0.5 (0.3) | −0.38 (−1.62 to 0.93) | 0.85 |

Note: * indicates significant ($p < 0.05$) differences in blocked shots between M-types and N-types.

## 3. Discussion

This is the first study to examine the effect of chronotype on in-game performance among professional basketball players with previous research either exploring the relationship between chronotype and in-game baseball performance [17] or diurnal variations in sport-specific skills among athletes competing in soccer [1,12], badminton [11], and tennis [9,10]. The main findings contrast the hypothesis, showing no differences in in-game performance during games commencing >18:00 h despite effect size analysis suggesting small to very large differences in performance measures between M-types and N-types.

Contrary to previous research establishing a time-of-day effect specific to chronotype for sport-specific skill performance [1,2,9,10,17], the current study did not observe a significant effect of chronotype on most in-game performance measures. It is plausible that the lack of effect regarding chronotype may have resulted from covariables, such as player roles and team tactics, suggesting that similar to other invasion sports, such as soccer, in-game basketball performance is influenced by other multifaceted variables that override any effect of chronotype [26–28]. That is, the potential effect of chronotype on in-game performance is masked by overall team performance in contrast to individual sports, whereby fluctuations in performance are the direct reflection of one athlete. For

example, a player's role within the team is not defined by their chronotype such that N-types do not assume being starters for matches played later in the day and become bench players in matches played earlier in the day. Instead, whether a player is a starter or bench player may reflect the established ability of each individual (e.g., higher scoring proficiency and higher rebounding effectiveness) compared to bench players [26]. While N-types received greater playing time compared to M-types in the current study, the lack of differences in the relative performance measures (i.e., offensive rating, defensive rating, and player efficiency) between groups suggest that N-types were able to sustain the same level of performance for longer, thereby implying that these players were of a higher level than players in the M-type group. The potential effect of player roles in team invasion sports relative to individual sports or team sports comprising skills primarily reflective of fluctuations of one player's performance may explain why an effect of chronotype or diurnal variation has been observed for badminton serve accuracy [11], baseball batting average [17], soccer dribbling [1], soccer chipping and volleying tasks [1,2], and tennis serve accuracy [9,10] but not in the current study.

Alternatively, player roles within a team may reflect the team strategy for each particular game. Indeed, Clemente et al. [29] and Mexas et al. [30] observed individual performance in basketball to be reflective of the player's role. Point guards were found to be the most prominent position during offensive organization [29], while perimeter players were observed to be primarily responsible for the majority of offensive efforts relative to post-players [30]. Post-players in comparison were observed to be the least prominent position during offensive organization with their primary role to receive the ball and shoot [29]. The influence of player roles on individual game performance potentially alludes to understanding the small to very large differences in in-game performance measures between chronotype groups. For example, the M-type group consisted of two centers and two guards (with most data for one guard excluded due to the fact of playing less than 10 min in most games), while the N-types consisted of four forwards and two guards. It may therefore be expected that due to the fact of being a taller group, M-types would record more blocked shots and rebounds in games. In contrast, the N-type group may be viewed as consisting of more adept shooting players or players prone to completing a higher number of assists and steals per game due to the positional requirements associated with the players in this group. These notions were reflected in the present findings with M-types registering significantly more blocked shots per game, while N-types exhibited more made and attempted three-point shots, assists, and steals per game. This supposition of a player's role affecting individual performance is supported by Sampaio et al. [31], who suggested that the prominence of each position on the court influences a player's involvement both offensively and defensively. Supporting the current data, Sampaio et al. [31] likewise observed perimeter players to be more likely to attain a higher number of assists and steals compared to post-players who were more likely to achieve a higher number of blocked shots. The potential cumulative effect of player roles and team tactics may therefore explain the lack of difference observed among most in-game performance measures between M-types and N-types.

The effect of habitual training time in accentuating differences between players who train and compete in the morning compared to those who train and compete in the evening is a factor that needs consideration when investigating the lack of differences observed between chronotype groups. In the present study, players consistently trained at 08:45–11:00 h across the season, while all games examined started >18:00 h. There is a consensus that acrophases, such as that of body temperature, can either be phase advanced or delayed thereby modulating an individual's chronotype based on the effect of exogenous and endogenous factors [32,33]. The ability to modulate chronotype is possible, as the heritability of chronotype is suggested to fall anywhere between ~25–50%, thus catering to the ability to shift the acrophases of psychobiological measures due to the presence of factors such as sunlight [34–37]. It is plausible that the early morning training sessions modulated the chronotype of the N-types to reflect that of M-types. That is, the early

morning training sessions may have phase advanced the acrophase of body temperature for N-types to earlier in the day, thereby providing no advantage to N-types over M-types during evening games. Rae et al. [38] supports this notion finding that ~70% of swimmers examined ($n$ = 26) performed better in time trials that aligned to the time they habitually trained irrespective of their chronotype. The plausible effect of habitual training time may encourage sports practitioners and coaching staff to schedule training in line with that of competition start time and highlights the importance for future research to examine the impact of different habitual training times on in-game performance while considering player chronotype. Such investigation will help determine if habitual training time impacts athletic performance according to player chronotype in basketball.

It is acknowledged that the key limitation of this study is the sample size distribution across chronotype groups and the exclusion of E-types. However, only a single professional basketball team was able to be recruited given the difficulties associated with recruiting multiple professional teams from the same league. The single-team recruitment therefore limited the number of definite M-type and E-type players able to be included.

A second limitation of this study is that only male players were included due to the fact of recruiting players from the same professional team. Therefore, the current findings are likely to not be representative of female players or semi-professional and amateur players who may be subject to other contextual factors (e.g., work commitments to supplement their living and different training times) that may affect their circadian rhythm or present as different chronotypes [39,40]. However, the sample of this study reflects similar chronotype studies examining high-level athletes given the limited M-types and E-types prevalent within these types of athletic samples [41,42]. It is important for future studies to examine the effect of chronotype on sport-specific performance during games among female players and players of different competition levels to understand how chronotype may influence performance specifically in these populations.

*Practical Applications*

Player chronotype did not affect in-game performance in professional male basketball players with performance measures remaining consistent between M-types and N-types in evening games played after 18:00 h. From a practical perspective, basketball coaching staff may not need to consider player chronotype when developing match preparation strategies or assigning starters and bench players when playing in the evening if their team largely consists of M-types and N-types. However, coaches may endeavor to match habitual training times with that of games to ensure the greatest specificity and align player circadian rhythms to that of competition.

## 4. Methods and Materials

### 4.1. Participants

Professional basketball players ($n$ = 11) were recruited from the same team registered in the National Basketball League (NBL). The NBL is the leading professional basketball competition in Australia. Data were collected across all 20 rounds of the regular season and the first round of the finals during the 2019/20 NBL season. In total, the team competed in 31 games across the season between October 2019 and March 2020. The regular season included 12 single-game weeks and 8 double-game weeks, while the first round of the finals (played across 2 weeks) included 1 single-game week and 1 double-game week. The team competed in 7 games at <18:00 h (15:00 h—1; 16:00 h—2; 16:30 h—1; 17:00 h—1; 17:30 h—1) and 24 games at >18:00 h (18:30 h—10; 19:00 h—7; 19:30 h—7). Across the season, including finals, the team had 17 wins and 14 losses. All playing positions were represented among the players including guards ($n$ = 5), forwards ($n$ = 4), and centers ($n$ = 2). Guards recorded 118 data samples (Player 1: 2; Player 3: 26; Player 5: 30; Player 7: 29; Player 9: 31), 67 forwards (Player 4: 31; Player 6: 13; Player 8: 2; Player 11: 21) and 58 centers (Player 2: 27; Player 10: 31). Descriptive statistics for each chronotype group are presented in Table 2. Players were screened for any sleep disorders using the

Pittsburgh Sleep Quality Index (global sleep quality index: 1-7; [43]) prior to the study's commencement. Each player provided written informed consent and were healthy without any injury or illness. The study was approved (14 July 2020) by the CQUniversity Human Research Ethics Committee (no: 21175).

**Table 2.** Median (inter-quartile range) characteristics of the professional male basketball players recruited in this study.

| Characteristic | Chronotype Group | | All Players |
| --- | --- | --- | --- |
| | M-Types (*n* = 4) | N-Types (*n* = 6) | |
| Age (y) | 28.5 (24.5–32.5) | 24.0 (22.5–25.5) | 24.5 (23.5–26.8) |
| Height (cm) | 195.5 (187.5–204.8) | 201.5 (195.0–202.8) | 201.5 (193.0–203.0) |
| Body mass (kg) | 95.5 (85.3–111.3) | 96.0 (93.3–99.5) | 96.0 (87.8–103.8) |
| National playing experience (y) | 8.0 (5.3–11.3) | 4.0 (2.5–4.8) | 4.5 (3.3–6.8) |
| MEQ score | 62.5 (59.8–67.5) | 51.5 (48.8–54.3) | 56.5 (51.3–59.8) |

E-types were excluded due to the lack of representation in the sample (*n* = 1). MEQ = morningness–eveningness questionnaire.

### 4.2. Morningness–Eveningness Questionnaire

The MEQ is a 19-item questionnaire used to determine when the respondent feels most inclined to complete certain behaviors over a 24 h daily cycle [44]. A value is assigned to each response with the sum of scores ranging between 16 and 86 [44]. A range of values is designated to each chronotype from the sum of scores to establish respondent chronotypes such that M-types reflect scores ranging between 59 and 86, N-types between 42 and 58, and E-types between 16 and 41 [44]. Out of the 11 basketball players, four self-reported as M-types (MEQ > 59; 86 data samples), six self-reported as N-types (MEQ 42–58; 126 data samples), and one self-reported as an E-type (MEQ < 41; 31 data samples). Given the lack of E-types, the initial hypothesis could not be tested, and an updated hypothesis was developed. The revised aim of this study was to identify whether there was a difference in in-game performance between N-types and M-types during evening games (>18:00 h).

### 4.3. In-Game Performance Measures

In-game performance was determined using game-related statistics. These statistics were recorded by qualified personnel and were freely available online (nbl.com.au; retrieved on 10 April 2020) following each game. Game-related statistics were imported into a Microsoft Excel spreadsheet (Version 15.0; Microsoft Corporation, Redmond, WA, USA) for further calculations. Performance was determined for each player by using individual statistics as summarized in Table 3. In addition, composite measures were used to indicate overall player performance including effective field goal percentage, offensive rating (points + rebounds + assists + steals + blocks), defensive rating (missed field goals + missed free-throws + turnovers), and player efficiency (offensive rating–defensive rating). These measures have previously been used in basketball studies to indicate in-game player performance [21,45].

### 4.4. Statistical Analysis

Game data were only included if players participated in more than 10 min of live time during gameplay to ensure sufficient participation was registered [46]. Due to the fact that only one player identified as an E-type in the recruited sample, E-type data were excluded from the analysis. The Shapiro–Wilk test demonstrated that the data were not normally distributed ($p < 0.05$). Separate linear mixed models were conducted to determine the effect of chronotype on each in-game performance measure. To investigate the impact of chronotype on each performance measure, each mixed model included chronotype as a fixed effect and player as a random effect. Effect sizes with 95% confidence intervals were also calculated to quantify the magnitude of difference in each in-game performance measure between chronotypes with the effect magnitude interpreted as: <0.20 (trivial),

0.20–0.59 (small), 0.60–1.19 (moderate), 1.20–1.99 (large), and >2.0 (very large) [47]. All statistical analyses were performed using SPSS statistics (Version 25, IBM Corporation; Armonk, NY, USA). Post hoc power analyses were performed using G*Power (Version 3.1.9.4, HHU, Düsseldorf, Germany). Player characteristics and in-game performance measures are expressed as the median and inter-quartile range (IQR) with statistical significance for all analyses set at $p < 0.05$. Post hoc power analyses are expressed as absolute numbers with statistical power set at 0.80. In addition, the mean and standard deviation were calculated for ease of interpretating in-game performance measures comparatively.

**Table 3.** In-game performance measures used in this study and their associated definitions [21,45].

| Measure | Definition |
| --- | --- |
| Minutes | Total playing time |
| 2-points made | Total number of successful 2 point shots during the game |
| 2-points attempted | Total number of attempted 2 point field goals during the game |
| 3-points made | Total number of successful 3 point shots during the game |
| 3-points attempted | Total number of attempted 3 point field goals during the game |
| Points | Total points scored by a player during the game |
| Effective field goal percentage | $(FGM + 0.5 \times 3\ PM)/FGA$ |
| Assists | Total passes to a teammate that lead to a score |
| Rebounds | Total offensive and defensive rebounds for a player during the game |
| Steals | Number of times a player legally causes a turnover defensively |
| Blocks | Number of times a player legally deflects an opponent's shot defensively |
| Offensive rating | Positive contributions made to the game (points + rebounds + assists + steals + blocks) |
| Defensive rating | Negative contributions made to the game (missed field goals + missed free-throws + turnovers) |
| Player efficiency | Offensive rating–defensive rating |

## 5. Conclusions

There was no definitive effect of chronotype between M-types and N-types on in-game performance in evening games (>18:00 h) among a professional male basketball team despite small to very large effects in blocked shots favoring M-types, as well as made three-point shots, attempted three-point shots, assists, and steals favoring N-types. Further research examining differences in in-game performance during games commencing earlier in the day (<18:00 h) compared to those in the evening (>18:00 h) and the interrelating effects of habitual training time on in-game performance according to chronotype is encouraged in basketball players.

**Author Contributions:** Conceptualization, M.P. and M.L.; methodology, M.P., M.L., N.E., J.G. and A.S.; software, M.L.; validation, N.E., J.G. and A.S.; formal analysis, M.P. and M.L.; investigation, M.P.; resources, M.L. and N.E.; data curation, M.L. and N.E.; writing—original draft preparation, M.P.; writing—review and editing, M.P., M.L., A.S., N.E. and J.G.; visualization, M.P., M.L., A.S., and N.E.; supervision, M.L.; project administration, M.P. All authors have read and agreed to the published version of the manuscript.

**Funding:** This research received no external funding.

**Institutional Review Board Statement:** The study was conducted according to the guidelines of the Declaration of Helsinki and approved (14 July 2020) by the Institutional Review Board (or Ethics Committee) of CQUniversity (no: 21175).

**Informed Consent Statement:** Informed consent was obtained from all subjects involved in the study. Written informed consent was obtained from the patients to publish this paper.

**Data Availability Statement:** Data available on request.

**Acknowledgments:** The authors would like to express their gratitude to the players and coaching staff for their contributions in completing this study.

**Conflicts of Interest:** The authors declare no conflict of interest.

## Appendix A

**Table A1.** Post hoc power analyses (two-tailed and $\alpha = 0.05$) for comparisons of in-game performance measures between morning-types (M-types) and neither-types (N-types) during evening games in professional male basketball players.

| Measure | Cohen's d (95% CIs) | Power | Required Sample Size (Power of 0.80) | | |
|---|---|---|---|---|---|
| | | | **M-Types** | **N-Types** | **Total Sample** |
| Minutes | −0.95 (−2.18 to 0.46) | 0.24 | 17 | 25 | 42 |
| 2-Points made | 0.28 (−1.01 to 1.53) | 0.07 | 176 | 264 | 440 |
| 2-Points attempted | 0.36 (−0.95 to 1.60) | 0.08 | 107 | 161 | 268 |
| 3-Points made | −1.98 (−3.28 to −0.30) | 0.74 | 5 | 7 | 12 |
| 3-Points attempted | −1.86 (−3.14 to −0.21) | 0.69 | 6 | 8 | 14 |
| Points | −0.77 (−2.00 to 0.60) | 0.18 | 24 | 36 | 60 |
| Effective field goal percentage | −0.38 (−1.62 to 0.93) | 0.08 | 96 | 144 | 240 |
| Assists | −1.42 (−2.67 to 0.10) | 0.47 | 8 | 12 | 20 |
| Rebounds | 0.74 (−0.63 to 1.97) | 0.17 | 26 | 40 | 66 |
| Steals | −3.03 (−4.48 to −0.99) | 0.98 | 3 | 5 | 8 |
| Blocks | 0.4 (−0.91 to 1.64) | 0.08 | 87 | 131 | 218 |
| Offensive rating per minute | −0.29 (−1.54 to 1.01) | 0.07 | 164 | 246 | 410 |
| Defensive rating per minute | 0.67 (−0.69 to 1.89) | 0.14 | 32 | 48 | 80 |
| Player efficiency per minute | −0.38 (−1.62 to 0.93) | 0.08 | 96 | 144 | 240 |

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
