# Peer review of "Player Chronotype Does Not Affect In-Game Performance during the Evening (>18:00 h) in Professional Male Basketball Players"

_2624-5175, doi:10.3390/clockssleep3040044_

Round 1
Reviewer 1 Report
Thank you for the opportunity to review the manuscript titled “Player chronotype does not affect in-game performance during the evening (>18:00h) in professional, male basketball players”, the authors have conducted a novel study investigating the effects on chronotype on professional male basketball players. However, some issues need to be addresed before consider for being published.
ABSTRACT
Line 12: “sport specific skills…” please provide examples.
Line 12: “across various team sports “please provide examples.
Lines 13-14: The authors stated “no research has explored the effects of athlete chronotype on in-game performance according to time of day across team sports”. The authors should consider changing the statement due to some previous studies have been realized in team sports such as field hockey (Facer-Childs et al 2015).
Reference:
Facer-Childs, E. and R. Brandstaetter, The impact of circadian phenotype and time since awakening on diurnal performance in athletes. Current Biology, 2015. 25(4): p. 518-22.
Lines 15-17: The authors stated “male basketball players (n = 11) but only reported, 10 chronotypes (morning-types: n = 4; neither-types: n = 6). Please clarify,
Line 20: Please provide exact p-value
INTRODUCTION
Line 35: “Daily circadian rhythmical oscillations” Please provide some examples.
Lines 50-52: The authors stated “Further, there remains limited data examining the effect of chronotype on in-game performance according to time of day. Please see (Vitale et al 2017) and rewording.
Reference:
Chronotype, Physical Activity, and Sport Performance: A Systematic Review. Vitale JA, Weydahl A. Sports Med. 2017 Sep;47(9):1859-1868. doi: 10.1007/s40279-017-0741-z.
Line 83-84: offensive rating (r = 8.59, p = 0.02) as well as player efficiency (r = 5.49, p = 0.01). Please clarify r values.
METHODS:
In this section the authors need to clarify some methodological issues. First, did the authors do sample size calculations for your measures of interest prior to running this study? If so, please present this data. If not, can you please run a post-hoc statistical power and sample size analysis? Please explain this point carefully.
Lines 114-115: Did the authors explain if basketball players have consumed caffeine prior the matches? Due to caffeine ingestion could impact on circadian rhythms effects (Mora Rodríguez 2012, 2015, Stojanović 2021),
References:
Caffeine ingestion reverses the circadian rhythm effects on neuromuscular performance in highly resistance-trained men. Mora-Rodríguez R, García Pallarés J, López-Samanes Á, Ortega JF, Fernández-Elías VE. PLoS One. 2012;7(4):e33807. doi: 10.1371/journal.pone.0033807. Epub 2012 Apr 4.
Improvements on neuromuscular performance with caffeine ingestion depend on the time-of-day. Mora-Rodríguez R, Pallarés JG, López-Gullón JM, López-Samanes Á, Fernández-Elías VE, Ortega JF.J Sci Med Sport. 2015 May;18(3):338-42. doi: 10.1016/j.jsams.2014.04.010. Epub 2014 Apr 26.
Acute caffeine supplementation improves jumping, sprinting, and change-of-direction performance in basketball players when ingested in the morning but not evening.
Stojanović E, Scanlan AT, Milanović Z, Fox JL, Stanković R, Dalbo VJ.Eur J Sport Sci. 2021 Feb 2:1-11. doi: 10.1080/17461391.2021.1874059. Online ahead of print
Reviewer 2 Report
Interesting study idea, my recommendations are the following:
In the participant section you mention 11 subjects and in table 1 and 3 there are only 10 (4 + 6). I recommend clarification.
Lines 136-137 repeat the aim. There is no revision compared to the one mentioned in lines 95-96. I recommend deleting. Repeat the aim again on lines 191-192.
